# More than Just Skin-Deep: A Review of Imaging’s Role in Guiding CAR T-Cell Therapy for Advanced Melanoma

**DOI:** 10.3390/diagnostics13050992

**Published:** 2023-03-05

**Authors:** Conor M. Prendergast, Kathleen M. Capaccione, Egesta Lopci, Jeeban P. Das, Alexander N. Shoushtari, Randy Yeh, Daniel Amin, Laurent Dercle, Dorine De Jong

**Affiliations:** 1Department of Radiology, Columbia University Irving Medical Center, New York, NY 10032, USA; 2Department of Nuclear Medicine, IRCSS Humanitas Research Hospital, 20089 Milan, Italy; 3Memorial Sloan Kettering Cancer Center, New York, NY 10065, USA

**Keywords:** melanoma, CAR T-cell, metastases, radiomics, PET radiotracers, CAR T-cell imaging, radiology, CAR T-cell toxicity

## Abstract

Advanced melanoma is one of the deadliest cancers, owing to its invasiveness and its propensity to develop resistance to therapy. Surgery remains the first-line treatment for early-stage tumors but is often not an option for advanced-stage melanoma. Chemotherapy carries a poor prognosis, and despite advances in targeted therapy, the cancer can develop resistance. CAR T-cell therapy has demonstrated great success against hematological cancers, and clinical trials are deploying it against advanced melanoma. Though melanoma remains a challenging disease to treat, radiology will play an increasing role in monitoring both the CAR T-cells and response to therapy. We review the current imaging techniques for advanced melanoma, as well as novel PET tracers and radiomics, in order to guide CAR T-cell therapy and manage potential adverse events.

## 1. Introduction

Worldwide, incidences of melanoma have been increasing faster than nearly all other cancers across all age groups. Typically, primary melanoma is detected through physical and dermatological examination; however, it is a notoriously aggressive cancer (the deadliest form of skin cancer), and its metastases can invade nearly every organ system and carry a poor prognosis. Therefore, radiology plays an increasingly valuable role in detecting, prognosticating, and managing later stages of the disease. We will provide the most comprehensive review thus far of medical imaging’s current and future role in advanced melanoma and discuss its expected role in guiding and monitoring CAR T-cell treatment for this disease.

### 1.1. Clinical Setting of Melanoma

As with most skin cancers, the main risk factor for melanoma is ultraviolet radiation exposure. Exposure to ultraviolet rays causes DNA damage to cells, which leads to cell-cycle arrest and apoptosis in normal cells, or abnormal proliferation and further mutation in cancer cells. Melanoma is currently ranked the sixth most common fatal cancer in the United States; this is expected to rise as population densities increase near the equator and with more frequent use of indoor tanning beds and outdoor activities. Fortunately, as for other skin cancers, the risk for melanoma can be easily lowered by minimizing time in the sun, avoiding exposure during peak hours, and applying sunscreen.

Early-stage melanoma can be treated with surgery quite successfully. Detection and diagnosis occur via physical and dermatological examination, which has been supplemented by the invention of smartphone applications and spectrophotometric devices, which aid physicians in determining whether a lesion should be biopsied or excised [1]. Multiple guidelines for pathological analysis were developed throughout the 20th century, taking into consideration the strata of skin invaded by melanoma, the depth of invasion, and its spread to sentinel lymph nodes; these guidelines, while revised several times, have been used to determine treatment plans, including surgery, lymph node dissection, chemotherapy, and targeted therapy [1]. The latter options, discussed below, are utilized when melanoma has spread beyond the local lymph nodes and is not surgically resectable.

### 1.2. Current Limitations of Therapies for Advanced Melanoma

Melanoma outcomes are largely dependent on the stage of the disease at diagnosis. In the localized stage, the relative survival rate is approximately 98%. Where there is regional spread of the disease, the relative survival rate is 60–65%, and in distant melanoma the rate is only 18% [1]. Survival has markedly increased since the approval of immunotherapy and targeted therapy in 2011 [1]. The overall relative survival rate is now 93%, with 99% in localized disease, 70% for patients with regional spread of the disease, and 31% for patients with metastatic disease [1].

Prior to the advent of immunotherapy and targeted therapy, melanoma was treated using surgery and chemotherapy [2]. Surgical resection was the primary form of treatment for early-stage melanoma and could be curative [3]. Metastatic disease, however, had dismal surgical outcomes, with five-year survival rates less than 10% [4]. Given the poor response rates and its predominancy partial response rates, chemotherapy was considered palliative with eventual disease progression [5].

#### 1.2.1. Chemotherapy

The first chemotherapeutic drug for melanoma, imidazole carboxamide, or dacarbazine, was approved in 1975 [6]. This approval came following several clinical trials that tested the effectiveness of various chemotherapeutic drugs to treat melanoma. A trial in 1971 randomized 155 patients and found that dacarbazine yielded a 28% objective response rate in patients, making it the most effective single chemotherapeutic agent for melanoma that had been tested [7]. A subsequent study tested dacarbazine with carmustine and found no increase in efficacy in the group treated with both agents combined compared to dacarbazine alone [8].

Despite approval in 1975, researchers continued evaluating other chemotherapeutic options to improve response rates to therapy, including combination chemotherapy; however, these studies did not result in any agents that improved outcomes [9,10]. A trial in 1999 compared the Dartmouth regimen, which included dacarbazine, cisplatin, carmustine, and tamoxifen, to single-agent dacarbazine [10]. A total of 240 patients with Stage IV melanoma were randomized to either single-agent dacarbazine or the Dartmouth regimen; however, results demonstrated that there was no significant difference in survival time or response between the groups [10]. While studies of alternate therapies did not result in an agent that could improve outcomes, researchers were able to identify a regimen of cisplatin and carboplatin that resulted in a response rate of 26.4% in patients with melanoma resistant to dacarbazine, providing a potential alternate therapeutic option [11].

#### 1.2.2. Surgery

Surgical procedures for localized disease offer potential for a cure and include wide excision and Mohs surgery, which completely remove the tumor [12]. Mohs surgery is a technique that uses horizontal sectioning and color coding of specimens to remove the skin layer by layer while allowing 100% marginal examination and maintaining proper tissue orientation [12]. It is frequently used in the head and neck region because it preserves more tissue than wide excisions [12,13].

Although excisions had been a longstanding treatment for melanoma, research was still being performed to improve it in the decades prior to the advent of immunotherapy and targeted therapy. In particular, researchers sought to find the ideal excision margins to use in surgery. Wide excisions are those with a margin of 3 cm, while narrow excisions have a margin of only 1 cm [14]. A clinical trial from 1988 compared 307 melanoma patients who had a wide excision with 307 patients who had a narrow excision; it was found that both excisions were equally effective [14]. Those who received the narrow excision had a four-year overall survival rate of 96.8%, and those who received the wide excision had one of 96.0% [14]. A later clinical trial from 2004 further evaluated the effect of margin size by comparing 453 patients receiving a narrow excision with 447 receiving a wide excision [15]. While their data supported prior studies showing similar overall survival rates, they found that narrow excision was associated with greater risk in patients with a poor prognosis [15].

Other procedures commonly employed in melanoma are lymphadenectomy and node biopsy [16]. A lymphoscintigraphy is a procedure in which sentinel lymph nodes are identified using a radiotracer so they can be removed surgically [16]. Lymph nodes in the region draining the primary site of disease are removed in order to assess for the local spread of disease and determine whether more extensive resection and/or systemic therapy is warranted [17]. The removal of these lymph nodes not only improves the 10-year disease-free survival rates of patients but also improves the survival of patients with nodal metastases [17]. Further, the use of this technique to identify the sentinel lymph nodes can limit morbidity for the patient by allowing the lymph node chain to remain largely intact, facilitating distal drainage.

#### 1.2.3. Photodynamic Therapy

Photodynamic therapy is an alternative option to conventional treatment methods that can be used in specific situations and is especially suitable for patients who are unable to undergo invasive procedures [18]. It involves the administration of a photosensitizer to targeted cancer cells and is minimally invasive [18]. After sensitization, light irradiation excites the absorbance band of the photosensitizer, producing reactive oxygen species, which results in toxicity and eventual cell death [19]. Although phototherapy is usually ineffective against melanoma due to melanin pigmentation, which provides a defense mechanism, it can still be effective in some cases [19]. A clinical trial from 2004 using the photosensitizer chlorine (6) on 14 patients with isolated skin melanoma demonstrated successful disease regression in all cases [20].

#### 1.2.4. Targeted Therapy

Melanoma was one of the first solid tumors treated with immunotherapy, as well as other precision medicine techniques. There are now numerous FDA-approved companion diagnostic devices that can quickly detect specific melanoma mutation genotypes. Similar diagnostics are available for myeloid, lymphoid, and lung cancers, in order to employ personalized treatment [21]; this includes targeted therapies, agents designed to target specific driver mutations, and functions for different melanoma genotypes.

One of the most common mutations in primary melanoma inhibitors is BRAF mutation, found in 50–60% of metastatic cases; thus, some of the first targeted agents to be developed were BRAF inhibitors [21]. Since 2011, the approval of the BRAF inhibitor vemurafenib, followed by dabrafenib and encorafenib, has improved outcomes and extended overall survival rates from 28% to 62% in early-stage disease [22].

The main cause of resistance to BRAF inhibitors (BRAFis) is activation of the MAPK pathway, an enzyme downstream of BRAF. It is reactivated through overexpression of NRAS, which is further upstream from BRAF and independent of BRAF activity. The BRAF protein may also overcome inhibitors via alternate splicing or overexpression of the BRAF allele [22]. Regardless of the mechanism, BRAFi resistance develops in approximately 20% of melanoma cases [22]. Potential agents to counteract BRAFis are being investigated. One early strategy attempted to target heat shock protein 90, which stabilizes and chaperones BRAF [22]. Efforts instead have focused on MAPK/BRAF combination inhibitors, including dabrafenib and trametinib, vemurafenib and cobimetinib, and encorafenib and binimetinib [22].

## 2. CAR T-Cell Therapy Overview

### 2.1. Pre-Clinical Development

Chimeric antigen receptor (CAR) T-cell therapy was first envisioned in the 1980s. In that decade, the potential tumor infiltrating lymphocytes was recognized, Awareness of the therapy grew, and in the 1990s, when viral vector techniques were optimized the first T-cells were produced in 1993, although the outcomes were poor [23]. Subsequent engineering of costimulatory genes and specific HLA antigens allowed CAR T-cells to persist in 2002, with mouse models of cancer being successfully treated in 2003, proving the concept. Multiple iterations and improvements culminated in the first human trials in 2013 [24].

### 2.2. Preparation of CAR T-Cells

CAR T-cells have undergone numerous improvements throughout the 2010s, including additional costimulators, cells that secrete anti-cancer molecules, and utilization of CRISPR systems for bioengineering; however, the general process remains the same (Figure 1). It involves extraction of T-cells from the patient, and then transduction of activated T-cells with a viral vector to express the chimeric antigen receptor—the receptor used to target the cancer cells in the patient [25]. The CAR enables T-cell cytotoxicity to be directed against the cancer cells in an HLA-independent manner. The process can take up to six weeks, during which time patients undergo lymphodepletion chemotherapy to reduce the cancer burden [25]. Then, the modified T-cells are transfused back into the patient [26].

Multiple generations of CAR T-cells have been produced since their introduction in 2002, mainly classified by the number of their intracellular signaling molecules. The first generation of CAR T-cells had only one domain, CD3ζ [25]. Subsequent second (currently the only FDA-approved) and third generations incorporated costimulatory domains for better antigen recognition and CAR T-cell activation, in order to improve efficacy and persistence [25]. The fourth generation emerged in 2015, which is equipped with further costimulatory molecules and also releases IL-12 to prime local innate immune cells, further helping to eliminate cancer cells [23]. Finally, the fifth generation of CAR T-cells is under development; they are equipped with “logic” domains, which are multiple receptors that confer on/off functionality when the cells bind, giving them greater specificity and reducing tumor evasion [27].

### 2.3. Current Clinical Application and Ongoing Studies

Currently, there are several FDA-approved CAR T-cell therapies for use in the United States for hematological cancers, including acute lymphoblastic leukemia (ALL) and diffuse large B-cell lymphoma (DLBCL). Efficacy data for both cancers treated with CAR T-cell therapy is promising, with a 70% response rate and 50% complete remission in DLBCL, and 93% complete remission in ALL patients [28].

CAR T-cell applications for advanced melanoma continue to be an area of interest, given conventional treatments’ poorer prognoses. Clinical trials are underway with novel GPA-TriMAR T-cells, essentially CAR T-cells bioengineered with lentivirus vectors against neoepitopes, or antigens unique to the patients’ particular melanoma gp100 peptides [28]. Others seek to combine the traditional chemotherapy agents cyclophosphamide and fludarabine with anti-VEGF CAR CD8+ cells in the hope of inducing synergy against metastases [29]. CAR T-cells are also being tested against recurrent, relapsing melanoma in the hopes of duplicating the treatment’s success story with hematological cancers [30].

### 2.4. Clinical Challenges

Naturally, there are limitations to this new method of treatment, regardless of the type of cancer being treated. Because CAR T-cells are highly specific to a particular tumor’s antigen, the treatment is prone to antigen escape, where the cancer cells discontinue expressing the target antigen. CAR T-cells exhibit reduced penetrance of solid tumors due to the stroma, which provides both a biochemical and physical immunosuppressive barrier. The target ligands of solid tumors can also be expressed on healthy tissue, presenting another obstacle [31].

The significant clinical challenge of imaging CAR T-cells during treatment for advanced melanoma and other cancers is the inability to effectively resolve the cells themselves. Peripheral blood draws only provide quantitative data, and reporter genes engineered into the therapeutic cells hinder the development of a universal CAR T-cell line [32]. Radiolabeled monoclonal antibodies appear to be an attractive solution for their specificity and temporal independence; however, human trials have not yielded appreciable in vivo data, and introduction of antibodies into CAR T-cell therapy could pose the risk of interfering with the efficacy of the treatment and potentially triggering an IRAE [32].

CAR T-cell therapy is not without its adverse effects. The most common reaction is cytokine release syndrome (CRS), an elevation in cytokines that induces systemic inflammation [33]. It is associated with life-threatening conditions such as renal insufficiency, cytopenia, hypotension, and coagulopathy. CAR T-cell therapy-associated neurotoxicity is also documented, referred to as immune effector cell-associated neurotoxicity syndrome (ICANS); it manifests as encephalopathy, though the mechanism is not fully understood [29]. CAR T-cell therapy also poses toxicity to other organ systems, detailed later.

### 2.5. A Promising Future

Despite its limitations, the development and deployment of CAR T-cells is an extraordinary achievement that holds tremendous promise for the future of immunotherapy. The ultimate, ambitious goal is to create a universal CAR T-cell (UCAR). UCAR T-cells would be taken from a healthy donor and transfected with a viral vector to engineer the specific CAR. They would not need to be harvested from the patient receiving treatment and could therefore be mass-produced, driving down cost and yielding a more accessible therapy [27]. New delivery forms are being tested well, with the testing of a gel form of CAR T-cells being implemented in mouse models enhancing anti-tumor activity [34]. CAR engineering does not stop with T-cells; CAR natural-killer cells and macrophages are also being investigated in clinical trials to expand the arsenal of immunotherapy agents against melanoma [35].

Several studies have investigated potential melanoma antigens for CAR engineering, including CD126, CD16, and HER2; CAR T-cells against these antigens have shown potent anti-tumor effects [28]. Clinical studies are in progress, but at the time of writing, most are either recruiting patients or have not published the results of this work. Still, with the success story against hematological cancers, there is great hope for future development of CAR T-cell therapies for advanced melanoma.

## 3. PET Imaging Techniques for Patients with Advanced Melanoma

### Cutaneous versus Uveal Melanoma

While most people think of melanoma as a skin disease, it can occur in any location where melanocytes are present. An important subtype is uveal melanoma (UM), sometimes referred to as ocular melanoma. Both this and cutaneous melanoma stem from cancerous melanocytes: CM, as the name suggests, from melanocytes in the skin, and UM from melanocytes in the uvea (the iris, choroid, and ciliary body). There are other, even rarer, subtypes including lentigo maligna, amelanotic, and acral lentiginous melanoma. This review will focus on CM, with UM detailed briefly below.

The subtypes have several differences that influence their management. CM metastasizes via the lymph system, most commonly to the lungs, liver, and brain. UM spreads via the circulatory system, owing to the extensive vasculature of the eye, most commonly to the liver [36] (Figure 2)**.** Unlike CM, which is commonly detected via dermatological exam, primary UM sometimes presents with ocular symptoms when it invades the eye tissues, including blurred vision and seeing shadows [36].

Both are also genetically heterogeneous, with each subtype having several different driver mutations. UM metastases are particularly nefarious as they are very resistant to chemotherapy and radiation; they are also unaffected by BRAF inhibitors because the mutation is usually absent in UM [30]. Mortality rates have remained unchanged, but this may change with the breakthrough discoveries of UM-specific mutations, including *GNAQ*, *GNA11*, and *CYSLT1* [30]. Selective inhibitors of *GNAQ* and *CYSLTR2* are in Phase 2 clinical trials, and drugs used in CM treatment, such as trametinib, are also being explored in clinical trials to test for UM sensitivity. Tebentafusp, a monoclonal T-cell receptor immune-cell mobilizer, has shown great effect against metastatic UM, and received FDA approval for use in 2022 [37].

CAR T-cell research in ophthalmic cancer is limited, including in UM, as with most solid tumors. A transgenic mouse model of melanoma demonstrated complete tumor regression through HER2-guided CAR T-cells; HER2 was selected because it is the majority biomarker expressed by UM cells [37]. Early clinical trials are testing GD2 CAR T-cells against GD2-positive UM cells [37].

## 4. Current Standard of Care Imaging

Imaging has played an increasingly important role in the surveillance and management of melanoma, although physical examination remains the mainstay method of detecting primary melanoma and local metastases.

### 4.1. Ultrasound

Ultrasound imaging exhibits the highest sensitivity and specificity for local lymph node surveillance in patients with Stage I and II melanoma, especially clavicular and axillary lymph nodes where physical examination exhibits higher rates of false negatives. It is especially recommended for patients with positive sentinel lymph node biopsies, as well as patients with high-risk primary tumors who decline biopsies [38]. However, the long-term benefits for survival of ultrasound surveillance are not yet clear [38].

### 4.2. Lymphoscintigraphy

Lymphoscintigraphy is a nuclear medicine imaging technique used to assess the sentinel lymph node draining of a primary melanoma lesion [39]. After injection of ^99^Tc sulfur colloid, patients are imaged via a gamma camera to assess the lymph node where the radiotracer is first found, thus following the path of potential metastatic cells [40,41]. Following the radiotracer throughout the lymph channels provides a means to track the first lymph node that would be encountered by malignant cells—the sentinel lymph node [40,42].

A limitation of lymphoscintigraphy is that if there is more than one primary lesion, sentinel lymph node identification can only be performed for one at a time. If attempts are made to perform multiple procedures in a time period shorter than the ^99^Tc half-life, the sentinel lymph node will not be clearly identified [43]. Despite this, lymphoscintigraphy is an important procedure for patients undergoing surgery for melanoma and can provide valuable information to help surgeons limit morbidity and mortality.

### 4.3. PET/CT

18-fluorodeoxyglucose positron emission tomography (^18^FDG-PET) combined with computed tomography (CT) (PET/CT) has been the favored modality for distant melanoma metastasis. In a meta-analysis of about 10,500 patients, PET/CT yielded accurate detection of metastases, with a sensitivity and specificity of 86% and 91% for distant metastases in patients with Stage IIB and III melanoma [21]. PET/CT is also useful in following up with asymptomatic patients after initial treatment, though cohort studies indicate they can show false-positive results in 9–14% of cases [21].

One of the most common sites of distant melanoma metastasis is the central nervous system (CNS), with tumors being detected in almost 50% of cases of Stage IV disease [37]. PET/CT is less useful in CNS detection owing to the high baseline metabolism of the brain resulting in high FDG uptake and lack of contrast on the CT. Therefore, magnetic resonance imaging (MRI) is superior for CNS metastasis detection because of its higher resolution. In particular, contrast-enhanced T1-weighting was found to be the most sensitive MRI sequence for detection of metastasis [43].

## 5. New and Emerging Imaging Modalities

^18^FDG-PET remains the “traditional” imaging for many cancers; though it is not specific to melanoma, this type of cancer is intensely ^18^FDG-avid, making it a very useful modality [44]. However, ^18^FDG-PET has a limited ability to detect the subcentimeter micrometastases of melanoma, and it is nonspecific. This had led to the exploration of novel biomarkers and imaging techniques with more sensitivity and specificity to melanoma cells, detailed below and compiled in Table 1.

### 5.1. PD-1/PD-L1 Targeting

PD-1 and PD-L1 are both immunotherapeutic monoclonal antibody checkpoint inhibitors that are approved by the FDA and EMA for multiple types of advanced cancer, including melanoma. While 40% of patients exhibit a response, no reliable biomarkers are available to predict responders [44]. However, the PD-1/PD-L1 monoclonal antibodies themselves can be radiolabeled for use in imaging, though their long clearance time necessitates the use of radionuclides with longer half-lives.

Murine models have tested checkpoint inhibitors tagged with ^89^Zr, ^64^Cu, and ^68^Ga. ^89^Zr was able to detect low levels of PD-L1 and evaluate changes in its expression in non-small-cell lung cancer xenografts [45]. ^64^Cu was less favorable versus ^68^Ga and ^18^F isotopes because of its very long half-life, as it would necessitate longer, unfavorable delays between injections. ^68^Ga performed well in mouse melanoma cells and quantified PD-L1 changes in response to chemotherapy [46]. It has also been used in Phase I clinical trials, where the radiolabeled peptide ^68^Ga-NOTA-WL12 uptake was strongly correlated with PD-L1-expressing non-small-cell lung cancer cells; it was also tolerated well by patients and cleared safely [46]. Theoretically, PD-1/PD-L1 PET imaging appears to be a bright prospect for monitoring tumor response to immunotherapy and CAR T-cell therapy, and with multiple larger-scale clinical trials underway, including in advanced melanoma, evidence may emerge to support it as a first-line imaging study.

### 5.2. Fibroblast Activation Protein (FAP) Targeting

Many cancers are associated with fibroblast cells in order to produce their tumor microenvironment and stroma. FAP is overexpressed by both cancer cells and their fibroblasts, making it a specific radiotracer as well. In preclinical studies it has shown remarkable uptake in rare tumors, both solid and hematologic, as well as their metastases. Multiple clinical trials are currently underway to evaluate FAP-targeted imaging tracers, which are evolving to increase tumor retention time and uptake, and are even more tumor-specific [47].

### 5.3. Melanin Targeting

Melanin is specific to melanoma cells and is often overexpressed in melanoma, making it an inherently attractive target for increasing tumor–background contrast on imaging. In mouse models and small cohorts of human subjects, it demonstrates superior uptake and detection of subcentimeter lung metastases when compared to ^18^FDG, which has been corroborated through human trials, with superior detection of microlesions without adverse effects [48].

Additionally, because melanin expression is a predictor of melanoma response to therapy, it can also serve as a probe for how effective a treatment could be in a particular patient. For example, a study showed that tyrosine, the amino acid precursor to melanin, promoted a phenotypic change that was associated with MAPK inhibitors [48]. This also implies that targeting melanin in PET imaging may complicate the clinical use of inhibitor therapy and should be an area of focus should the radiotracer enter clinical trials.

### 5.4. Benzamide Targeting

Benzamine and its derivatives are molecules with a strong affinity for melanin, making them an ideal choice for melanocyte-specific imaging. Iodinated, aminated, and alkylated derivatives have been explored in studies and clinical trials, though it was found that ^18^FDG tracers offered better sensitivity [46]. Subsequent preclinical studies with methoxylated benzamides presented a higher contrast between the tumor and background, owing to their low accumulate in normal tissue [49,50]. Lastly, while benzamide derivatives can be synthesized quickly and bind their targets strongly, they are slow to be metabolized and excreted, leading to retention and poorer detection of liver and gastrointestinal lesions [51].

### 5.5. Nicotinamide Targeting

Nicotinamide is also a melanin-specific agent via its use as a cofactor by tryptophan metabolism. Though it has yet to progress to clinical trials, nicotinamide radiotracers offer the advantages of early tumor uptake, being renally excreted due to hydrophilic pyridine, and quick synthesis via direct halogenation [50]. In mouse models, nicotinamide conjugated with benzamide derivatives demonstrated rapid and strong uptake by melanoma cells, and yielded a high contrast between the tumor and muscle [51]. While there appears to be potential, it has yet to be brought to human patients for targeted PET/CT use.

### 5.6. Integrin Targeting

Integrin is a transmembrane receptor of vascular endothelial cells. αvβ3 integrin is a player in angiogenesis in tumors, and with melanoma in particular, it promotes growth into the basement membrane for metastasis [52]. It is selectively expressed in abundance in melanoma tumors, and specific peptides have been bioengineered to exploit this property for PET imaging. Clinical trials with an ^18^F-Galacto-RGD (arginine-glycine-aspartate) PET tracer showed good biodistribution and αvβ3 integrin receptor binding [53]. Research remains in progress, and there is hope that improvements in integrin targeting could provide valuable information about tumor angiogenesis.

### 5.7. MEK Targeting

MEK is a protein directly downstream of BRAF, which, in more than 50% of melanoma cases, is constituently active [54]. This discovery led to the development of multiple MEK inhibitors, including trametinib. Its many side effects and associated toxicities spurred interest in developing a radioprobe to better identify patients with the mutant pathway who, therefore, benefit most from MEK targeting [55]. Trials with ^121^I-trametinib used in PET imaging on melanoma cancer lines found the tracer was indeed taken up significantly more by BRAF and KRAS mutant cells than wild types [56]. More studies must be performed to assess ^121^I-trametinib–PET’s potential in melanoma imaging before it can be approved for clinical use [57].

### 5.8. Imaging CAR T-Cells with PET Tracers

Once CAR T-cells are infused back into the patient, monitoring and tracking of them is traditionally performed via peripheral blood quantification. This method does not provide any insight into the cells’ bioavailability nor their activation status within the tissue [58]. Molecular imaging of CAR T-cells via an engineered reporter gene may be used, but it requires the creation of new cell lines that incorporate the imaging gene. To bypass this major limitation, radioactive CAR T-cell-targeting radiotracers are being tested [58].

External radiotracers may be applied to any CAR T-cell line, but have shorter half-lives and limited temporal resolution [58]. Radiolabeled antibodies are being explored instead, as monoclonal antibodies with radioisotopes offer the specificity required for CAR T-cell detection but can be injected at different timepoints following infusion [58]. Studies have investigated CD278, a costimulator upregulated in T-cell activation, as a target for monitoring CAR T-cell activity; in mice models, ^89^Zn-labeled antibodies against the costimulator outperformed controls [58]. More work must be undertaken to evaluate the potential for immune-response adverse effects (IRAE) or interference with the CAR T-cell therapy before translation to human trials.

### 5.9. Radiomics

Given the role of imaging in advanced melanoma management, it makes sense that the growing field of radiomics should also be applied. In short, radiomics involves the extraction of quantitative data from imaging for more precise diagnosis, prognosis, and treatment of lesions. This provides the opportunity to go beyond images and elucidate potential biomarkers and build parameters for quantifying change [59].

Melanoma (and its metastasis) is one of the most heterogeneous tumors; radiomics could therefore provide a bevy of biomarkers, including angiogenesis/blood flow, glucose metabolism, and necrosis [59]. Of particular interest is the “virtual biopsy”, through which Shofty and colleagues applied machine-learning paradigms to brain MRIs. Their radiomics analysis was able to predict BRAF status in CNS metastases, albeit with a mean accuracy of 79% [56]. Although not yet a refined tool, “virtual biopsy” and radiomics deserve future exploration due to their noninvasive and valuable nature, inexpensively provided.

## 6. Using Medical Imaging to Guide CAR T-Cell Therapy

CAR T-cell therapy has achieved remarkable therapeutic effects in cancer, particularly hematological cancer. However, the significant number of patients experiencing relapse and IRAEs during CAR T-cell therapy raises the need for reliable biomarkers for management, assessment of treatment response, and guidance of future care. Imaging via PET/CT already plays a foundational role in diagnosis and detection of melanoma and its metastases, but its quantitative parameters have proven clinically valuable as well, including SUV max, total metabolic tumor volume (TMTV), and total lesion glycolysis (TLG) [60]. Research into their value in the context of CAR T-cell therapy is new, but as it progresses, these parameters, and imaging overall, can be expected to more accurately quantify and monitor the treatment’s effect.

### 6.1. Response to Therapy and Monitoring

One of the overarching complications in assessing the response to CAR T-cell therapy, and perhaps immunotherapy as a whole, is reproducibility. As it is a new treatment, radiologists will need to stay up-to-date and learn the response patterns’ appearance in anatomic and metabolic imaging and better define the criteria for the atypical patterns observed.

### 6.2. Pseudoprogression

Another challenge of immunotherapy is differentiating progression of disease from pseudoprogression. This is defined as a perceived transient increase in tumor burden or size beyond the initial size, appearing shortly after administration of the immunotherapy agent [61,62,63]. Pseudoprogression is not true oncological progression and is soon followed by tumor regression. This occurs in nearly all solid tumors treated with immunotherapy, regardless of type. This is believed to be due to local immune response within the tumor microenvironment [61].

Conventional CT and MRI are used to identify pseudoprogression, but because its misclassification as true progression has exposed patients to unnecessary surgical risk, novel imaging techniques are needed [62]. PET/CT has proved useful in tumor staging and prognosis, and its semi-quantitative parameters (explored below) have gained interest for assessing pseudoprogression and typical response patterns.

To help standardize the assessment of tumor responses, the response evaluation criteria for solid tumors (RECIST) was introduced in 2000. It has since been revised as imaging and therapy has evolved, and novel response patterns have been identified as imaging has advanced. It provides a guideline for standardizing tumor changes based on lesion diameter, lesion burden, appearance/disappearance of lesions, and comparison to baseline FDG-PET uptake [64] (Figure 3 and Figure 4). The current guidelines consist of *complete response* (complete obliteration of lesions and regression of lymph nodal lesions <10 mm), *partial response* (30% decrease in sum of diameters), *stable disease* (no noteworthy changes), and *progressive disease* (20% increase in diameter sum from nadir) [64]. As advances in imaging better detect and standardize the pseudoprogression of the disease, we might expect it to also be added to RECIST guidelines in later evolutions.

### 6.3. SUV Max

Standardized uptake value is a dimensionless quantity in PET imaging. It is a ratio of tissue activity to an injected dose of activity and is a simple way to quantify the metabolic activity of a tissue in PET imaging. It is a dimensionless mathematical quantity, observer-independent, and the most commonly used parameter in clinical practice [64]. However, it can be influenced by many variables specific to the patient, including plasma glucose concentration, body size, and tumor type, and as well as the imaging instrument, such as noise ratio, imaging algorithm, and injection time [65]. Still, standardized uptake values provide strong outcome-predictive markers of tumor outcomes; higher SUVs are associated with poorer outcomes and greater recurrence rates in numerous cancers [65].

In the context of advanced melanoma, SUV max tended to be higher in non-survivors and patients with recurrence in a 2016 review [66]. However, it was found in a 2019 review that SUV max’s predictive power in initial scans before immunotherapy was not as high as scans obtained during treatment, such as after three and six months [65]. This dissonance emphasizes the need for standardization and reproducibility in quantifying responses.

### 6.4. Tumor Metabolic Volume

As the name suggests, tumor metabolic volume (TMV) is the volume of the tumor that is metabolically active on PET. Unlike SUV max, which is obtained from a voxel, TMV is more comprehensive and better reflects the total active tumor burden. Similar to SUV max, TMV demonstrates predictive value in monitoring immunotherapy; it shows strong negative predictive power at three and six months after initiation of treatment for advanced melanoma [65].

### 6.5. Total Lesion Glycolysis

Total lesion glycolysis is the product of TMV and average SUV. In 2019, retrospective cohort trials using FDG-PET imaging found TLG to be the best predictive biomarker in melanoma-specific survival, with a statistically significant relationship between TLG and overall survival when treated with immunotherapy [65].

### 6.6. Predictive AI Models

Early diagnosis of metastasis is paramount for management and prognosis. Artificial intelligence (AI) has been gaining traction with many industries in recent years, and it is expected that AI will aid diagnosis in future clinical practice. In oncology as a whole, there is great potential for AI to assist clinicians—improving detection, identifying and stratifying risk, monitoring response, and more.

In the context of melanoma, applied AI models have confirmed the validity of predictor variables of brain metastases, such as BRAF status, to predict response to treatment [63]. Additionally, a study of 448 melanoma patients utilized machine learning to define an algorithm for stratifying patients by risk of metastasis development; the algorithm correlated serum biomarkers (dermcidin and interleukin-4) with a greater probability of metastasis [63]. Similar studies have also quantified and identified fluctuations in cytokines to assess overall survival. Finally, AI can help clinicians monitor immunotherapy treatment outcomes beyond assessing for tumor reduction; radiomics signatures of tumors using CT can help estimate overall survival and differentiate response patterns earlier in treatment, helping to identify alternative treatment plans sooner, if necessary [67].

## 7. IRAE

Cytokine release syndrome (CRS) is one of the most common IRAEs associated with CAR T-cell therapy. This is a byproduct of their mechanism of action—upon binding to their target antigen, CAR T-cells proliferate and release cytokines, which can affect nearly every organ system. Common symptoms include increased vascular permeability, decreased cardiac output, fever, and hypotension [68]. Greater tumor burden appears to be a risk factor for more severe IRAEs; investigation into predictive models of potential biomarkers to stratify risk is ongoing and represents yet another useful realm of AI application [68]. 

CRS and IRAEs are typically managed with corticosteroids. Preliminary studies suggest that this does not interfere with CAR T-cell therapy response rates, but potential long-term effects are unknown [65]. In neurological events, patients should also receive brain imaging (preferably MRI) and appropriate neurological examination and should be frequently monitored until symptoms improve.

## 8. CAR T-Cell Toxicity

Numerous other CAR T-cell toxicities have been documented, many of them secondary to the physiological changes induced by CRS. Toxic effects can manifest in nearly every organ and can range from mild effects that physiologically compensate for CRS to severe, life-threatening conditions. They are briefly described here.

### 8.1. Cardiotoxicity

Cardiac side effects are hypothesized to be caused by CRS-induced capillary leakage or stress-induced cardiomyopathy [66]. They can present as tachycardia, arrhythmia, and heart failure, requiring critical care support. Management consists of cardiac monitoring following CAR T-cell infusion, and potential cardiac MRI and imaging to detect tissue changes [69].

### 8.2. Pneumonitis

This non-infectious process is a result of CAR T-cell cytokine release and must be promptly treated. It presents with nonspecific clinical and radiological signs indicative of alveolar damage, and can induce hypoxia in severe cases [67]. It is managed with anti-IL-6 agents, PD-1/PD-L1 and CTLA-4 inhibitors and supplemental oxygen, and mechanical ventilation in severe cases [70] (Figure 5).

### 8.3. Hepatomegaly

Rarely, CRS cytokines can induce hemophagocytic lymphohistiocytosis (HLH), a severe immune response that induces liver dysfunction [71]. It is diagnosed using CT, PET/CT, biopsy, and ultrasound. Care is mainly supportive, with steroid immunosuppressive agents in severe cases [71] (Figure 6).

### 8.4. Acute Kidney Injury

Cytokines from CRS events induce decreased cardiac output and vasodilation, leading to decreased renal perfusion and then necrosis, if under-perfusion persists [66]. Ultrasound is effective in identifying renal parenchymal damage, which may prompt a renal biopsy to assist with treatment options [72].

### 8.5. ICANS and Neurotoxicity

ICANS is the second most common CAR T-cell IRAE after CRS. Onset is most often four days after infusion, and includes a variety of neural symptoms, some severe: encephalopathy, headache, tremor, and aphasia. The onset of symptoms often overlaps with CRS [70]. MRI is the imaging modality of choice for evaluating ICANS due to its superior resolution to CT. Long-term follow-up is recommended to assess for potential cerebral damage [73] (Figure 7).

## 9. Conclusions

Owing to its success with hematological malignancies, research on applying CAR T-cell therapy to solid tumor research is booming, and there is reason to be optimistic about the future of CAR T-cell therapy for advanced melanoma. With its ability to deliver highly specific care tailored to each patient, CAR T-cell therapy could represent a great advance by bypassing the frequent development of the cancer’s resistance to conventional chemotherapy and provide a more targeted approach to distant metastases than surgery. In addition, potential for a universal therapeutic CAR T-cell line would include all of the mentioned benefits in an accessible and customizable treatment.

Medical imaging will continue to assist with the detection, screening, diagnosis, and prognosis of advanced melanoma, in addition to dermatological surveillance. To overcome the challenges of imaging both the therapeutic and cancerous cells during treatment, novel PET radiotracers are being developed, and radiomic approaches are being explored to offer more insight into tumor phenotypes. Imaging will play an important role in detecting and monitoring CAR T cell behavior and clinical response, to detect potential side effects toxicities, and guide clinicians in case of treatment failure. 

## Figures and Tables

**Figure 1 diagnostics-13-00992-f001:**
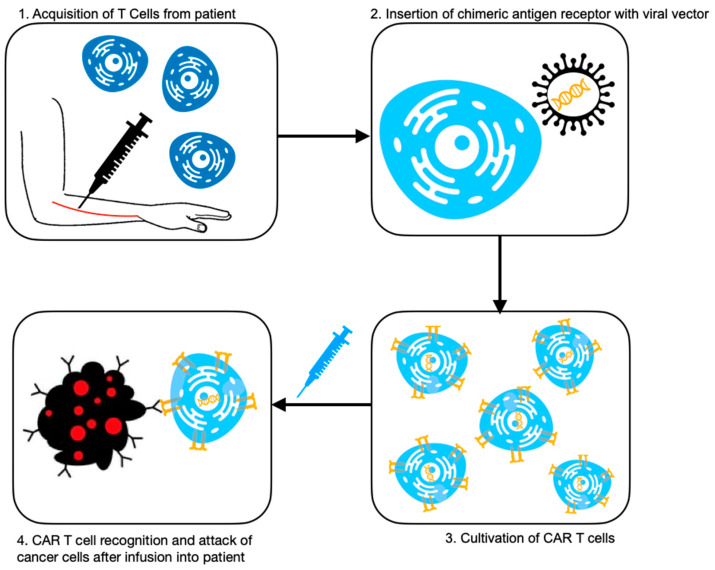
Summary of CAR T-cell therapy [23].

**Figure 2 diagnostics-13-00992-f002:**
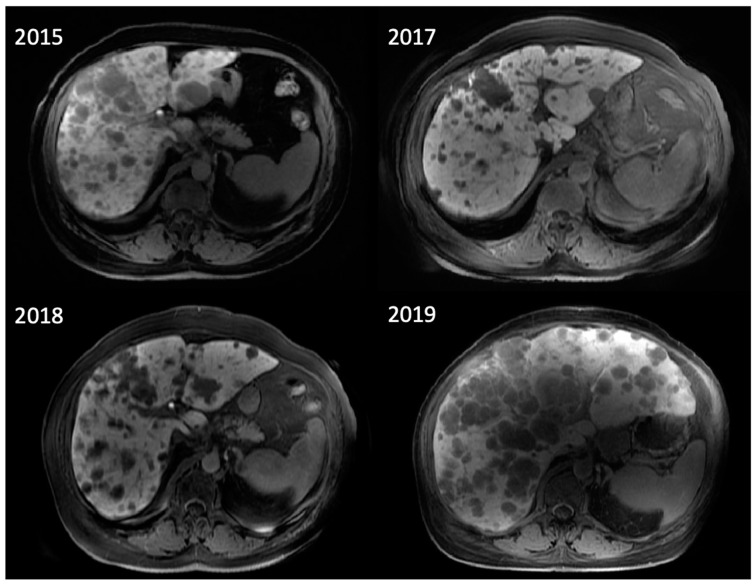
A 69-year-old woman with uveal melanoma metastatic to the liver post-treatment with tebentafusp, a bispecific antibody (BsAb), who developed cytokine release syndrome without radiologic imaging manifestations. Her large-burden liver disease was largely stable over the course of almost 3 years.

**Figure 3 diagnostics-13-00992-f003:**
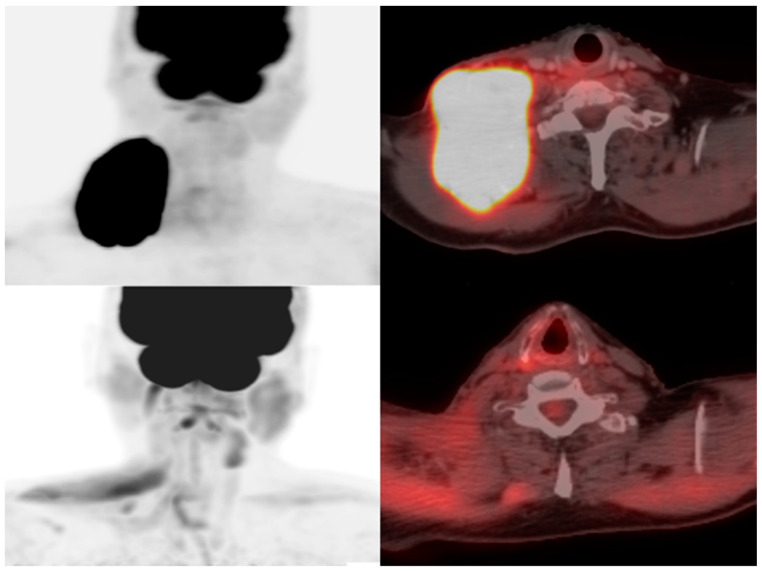
A 73-year-old man with diffuse large B-cell lymphoma. Maximum intensity projection and axial fused FDG PET/CT pre- (**above**) and 28 days post- (**below**) CAR T-cell therapy demonstrating resolution of hypermetabolic adenopathy in the right lower neck consistent with complete metabolic response.

**Figure 4 diagnostics-13-00992-f004:**
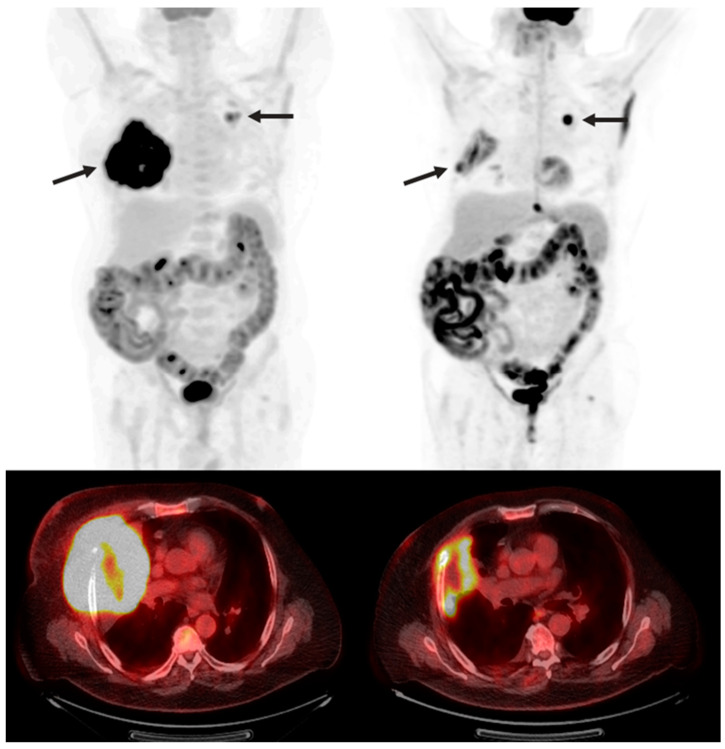
A 70-year-old man with refractory diffuse large B-cell lymphoma. Maximum intensity projection and axial fused FDG PET/CT pre- (**right**) and 3 months post- (**left**) CAR T-cell therapy showing marked decrease in size and FDG uptake of right upper lobe lymphomatous mass invading the right chest wall (arrow). Slight increase in FDG uptake of left upper lobe pulmonary nodule (arrow), also suspicious for lymphoma.

**Figure 5 diagnostics-13-00992-f005:**
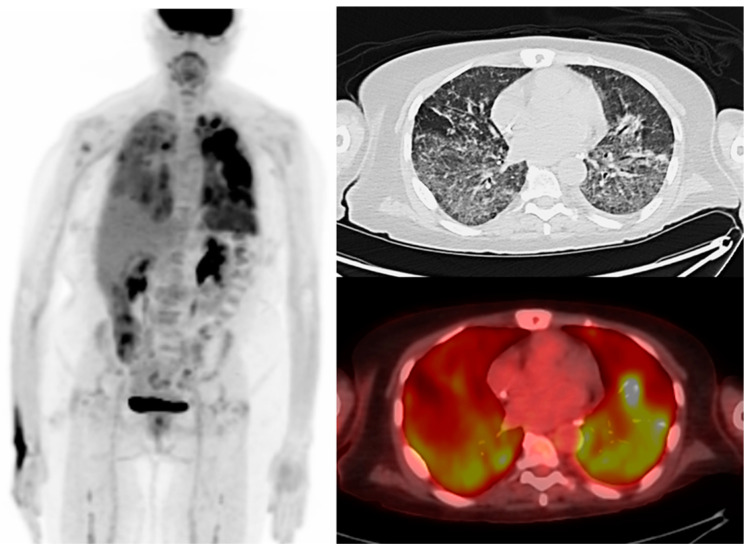
A 65-year-old man with relapsed multiple myeloma. Maximum intensity projection, axial CT, and fused PET/CT showing diffuse bilateral FDG-avid ground-glass pulmonary opacities and interlobular septal thickening, greater in the lower lobes on FDG PET/CT obtained post-CAR T-cell therapy suspicious for pneumonitis in the context of cytokine release syndrome.

**Figure 6 diagnostics-13-00992-f006:**
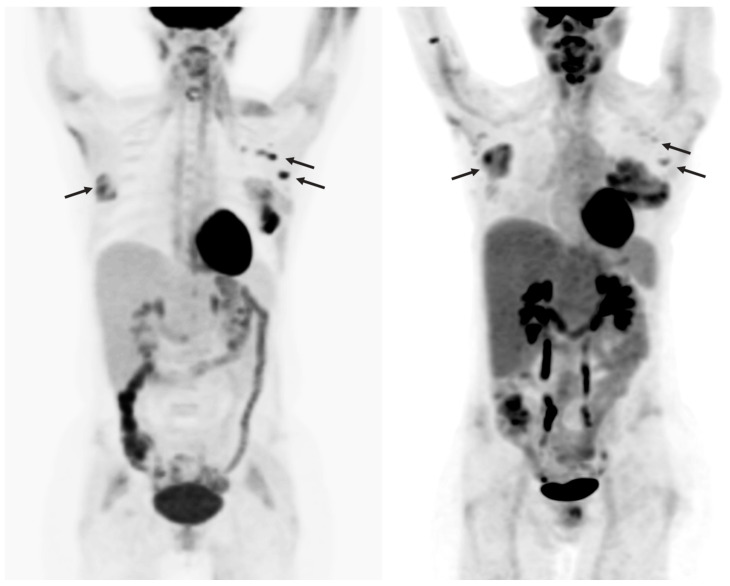
A 47-year-old woman with acute lymphoblastic leukemia and remote history of melanoma. Maximum intensity projection images pre- (**left**) and post- (**right**) CAR T-cell therapy showing new hepatomegaly with mild increased heterogeneous FDG uptake, with mixed changes in extramedullary leukemic involvement, increased in the right breast and decreased in the left axillary nodes.

**Figure 7 diagnostics-13-00992-f007:**
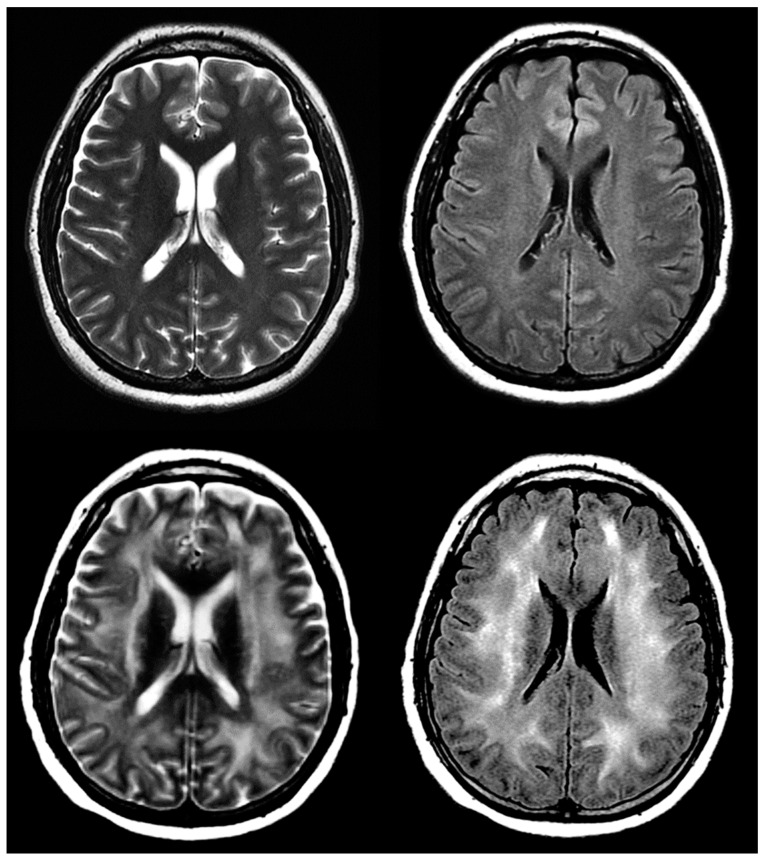
A 43-year-old woman with acute myeloid leukemia and history of confusion and disorientation 7 days post-treatment with CAR T-cells. Select images from axial T2 and FLAIR sequences pre- (**above**) and post- (**below**) CAR T-cell therapy demonstrating diffuse hyperintense signal abnormality in subcortical and supratentorial white matter, suspected for ICANS.

**Table 1 diagnostics-13-00992-t001:** Comparison of novel PET radiotracers with FDG-PET in melanoma models [38].

Tracer	Effectivity versus FDG-PET
PD-1/PD-L1: ^89^Zr-DFO-6E11 ^64^Cu-atezolizumab ^68^Ga-NOTA-Nb109	Detected the differences in PD-L1 expression of heterogeneous lymphoid tumors; tumor-to-muscle ratio correlated with tumor response to therapy Detected PD-L1 expression across various tumors; however, long delay between injection and imaging (24 and 48 h) is not clinically ideal Highly sensitive, but less specific than FDG-PET; monitored PD-L1 expression changes
FAP: ^68^Ga-FAPI-04 ^18^F-FAPI-04	Demonstrated high tumor to background ratio, particularly effective in rarer cancers; drawback is that it detects diseases with abundant fibroblasts besides cancer
Melanin: ^18^F-5-FPN ^18^F-DMPY2 ^18^F-ICF-1006	All had higher uptake vs. FDG-PET in B16F10 (mouse melanoma) cells; demonstrated superiority for visualizing subcentimeter lung metastases
Benzamides: ^123^I-BZA(2) 4-^11^C-MBZA ^18^F-MEL050	Exhibited comparable specificity vs. FDG-PET (79% vs. 94%), but significantly weaker sensitivity (39% vs. 87%). Superior to FDG-PET for higher tumor-to-background contrast ratio in melanoma Higher uptake in submillimeter lung metastases of mouse melanoma compared to FDG-PET
Nicotinamides: ^131^I-IFNABZA	Higher tumor-to-muscle ratio, and the advantage of renal secretion
Integrin: ^18^F-Galacto-RGD	Favorable specific receptor binding; was not compared to FDG-PET in studies reviewed
MEK: ^124^I-trametinib	More specific to melanoma cells; BRAF and KRAS mutants having a higher uptake than wild types; has not been approved for clinical trials

## Data Availability

Not applicable.

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
