# Peer review of "More than Just Skin-Deep: A Review of Imaging’s Role in Guiding CAR T-Cell Therapy for Advanced Melanoma"

_diagnostics, 2023, doi:10.3390/diagnostics13050992_

Round 1
Reviewer 1 Report
This review manuscript aimed at giving an overview of the current imaging techniques for advanced melanoma, as well as novel PET tracers and radiomics, in order to guide CAR T cell therapy and manage potential adverse events.
In the introduction part, the authors summarized the current limitations of therapies for advanced melanoma and gave an overview of current CAR T cell therapies. This is very helpful to readers to understand the background information. In addition, the author talked about cutaneous versus uveal melanoma. It further proved the necessity of imaging techniques in melanoma patients' treatment. Then, the authors summarized different imaging techniques and PET radiotracers with also their role in CAR T cell therapies.
At last, the authors pointed out CAR T cell therapy is quite hopeful for advanced melanoma. With novel PET radiotracers and radiomic approaches, more information can be revealed for the phenotype of advanced melanoma also for CAR T cell behavior in the patients. Such information can further help the development of CAR T cell therapies.
The language of the manuscript is clear and professional. However, the format of the titles of the paragraphs can be improved to show better the hierarchical structure of the review. For example, spacing in front of the paragraphs, highlighting some of the titles, etc.
Author Response
Thank you for you time and thoughts--we appreciate your feedback! The titles and paragraph spacing have been corrected to be more uniform and structured.
Reviewer 2 Report
The manuscript focuses on a systemic revision of literature data about the role of emerging CAR- T cell based approaches in the clinical administration of melanoma patients describes a timely relevant topci bus several integrations shjould be accepted to promote to publication of this mansucript on this journal
- In the introduction section, please could the authors detter describe the clinical setting of melanoma patients? Accordingly, major clinical and molecular integrations should be promtoed to improve the readibility of the manuscript
- In the study design section, I would strongly reccomend to review this section in order to clarify major points (topic, pre clinical application, clinical application, methodology, point of strenght and weakness) about CAR T in melanoma patients?
- Interestingly, could the authros focus on promising on going studies about this application?
- In the conclusion, please, could the authros emphasize the pivotal point approached by CAR T in comparison with standardized therapy?
Author Response
Thank you very much for your time and comments--we appreciate your feedback!
First, we expanded the introduction to include more background and clinical setting of melanoma.
Then, in the study design section, we added in more on the preclinical application and development of CAR T cells. We added in another paragraph detailing some the exciting clinical trials to further examine CAR T cell applications to metastatic melanoma.
Finally, we elaborated on CAR T cells' future role in the conclusion, and its potential advancement compared to conventional therapy.
Round 2
Reviewer 2 Report
The manuscript may be accepted in the present form